# Operational-Efficiency Improvement of Public R and D Management Agencies in South Korea

**Byung Yong Hwang [1],\*, Eun Song Bae [1], Heung Deug Hong [2] and Dae-cheol Kim [3]** 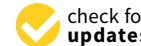

[1]  Korea Institute of S and T Evaluation and Planning, Seoul 06775, Korea; pearag@kistep.re.kr
[2]  Department of Administration, Kangwon University, Chuncheon 24341, Gangwon-do, Korea; hdhong@kangwon.ac.kr
[3]  School of Business, Hanyang University, Seoul 04763, Korea; dckim@hanyang.ac.kr
\*  Correspondence: byhwang@kistep.re.kr

**Abstract:** Public R and D management agencies have been taking on key roles in the national R and D ecosystem. The purpose of this study is to suggest ways to improve the operational efficiency of public R and D management agencies based on analysis of their current status. We approached this study from a life-cycle perspective as it applies to the plan-management evaluation of R and D. Data-collection sources included documents, surveys, and interviews with staff members in agencies responsible for national R and D management. Based on the analysis results, we present suggestions for improvement in three areas: (a) unification of R and D planning and evaluation of individual ministries; (b) establishment of a panministerial management system for public R and D management agencies; (c) improvement and development of public R and D agencies' expertise and management services. Finally, we discuss possible improvements and the limits of this study.

**Keywords:** national innovation system; public R and D management agency; life-cycle perspective; R and D performance

## 1. Introduction

With the diversification of government R and D entities and expanded investment in them since the establishment of the Korea Science and Engineering Foundation (KOSEF) in 1977, the number of R and D management agencies in South Korea (hereinafter Korea) has increased, along with the merging of some of these entities in 2008 [1] following public institutional reform. At present, 17 R and D management agencies are operating within 12 ministries/agencies. As we face the Fourth Industrial Revolution, there has been a growing awareness of the function and role of public R and D management agencies (hereinafter R and D management agencies) as the foundation for innovative growth and R and D efficiency.

R and D management agencies have been taking on key roles in the national R and D ecosystem. In addition, it is no exaggeration to say that the role of these institutions is directly related to the efficiency of government R and D, since these entities are implementing government R and D budgets at onsite R and D facilities. From this standpoint, it is natural that there is rising demand for the effective and efficient planning, management, and evaluation of such agencies at the national level in order to maximize the performance of government R and D.

However, there have also been concerns about a decline in the efficiency of government R and D due to the large number of operating R and D management agencies; some critics point out problems such as similar and overlapping R and D planning, insufficient ties between research results, and hindrances to researchers arising from disparate regulations, procedures, and the systems of the various agencies [2–4]. Thus, the Moon Jae-in administration has been pushing for the operational

efficiency of R and D management agenciesto overhaul the agencies and improve the work efficiency of researchers. To be more specific, the Moon Jae-in administration has been pushing ahead with "the operational efficiency of R and D management agencies and the foundation for public research" since this was selected as one of the Ministry of Science and ICT's national tasks in July 2017 [5]. This work aims to readjust, combine, and overhaul the functions of these disparate R and D management agencies by considering the characteristics of their policy targets and technological sectors. Moreover, national R and D innovation methods for the advancement of the National Innovation System (NIS) also contain details for stepping up government efforts to innovate R and D support systems by focusing on researchers, introducing R and D system and process innovations, and integrating R and D management agencies with the laws and regulations of overall R and D management [6].

The purpose of this study is to suggest various ways to support the operational efficiency of R and D management agencies based on analysis of the current status of these agencies. Specifically, this study focuses on three questions: First, what is the role of R and D management agencies under the National Innovation System (NIS)? Second, what is the governance system of R and D management agencies and their current status from an R and D life-cycle perspective? Third, how can the efficiency of R and D management agencies be improved?

## 2. Research Background and Methodology

### 2.1. National R and D Program Management Mechanism under the NIS

In the late 1980s, a new conceptual NIS framework was introduced by researchers, including Freeman [7], in science, technology, and innovation studies, and was later discussed by researchers including Lundvall [8] and Nelson [9].

The findings of recent studies are as follows. Jeon et al. [10] argue that the necessity of open innovation is magnified in NIS theory. Jang and Ko [11] tried to overcome the limit of macroscopic analysis levels of the NIS focused on creative-innovation cases at the individual and laboratory levels through participant observation of a public research institution in Korea. In addition, Hameed et al. [12] discussed the adaptability of the NIS for sustainable economic growth, looking at the example of South Korea's mired technology transfer and commercialization process, explaining that the NIS is a framework to study interactive learning and technological-capability accumulation processes in a nation, underpinning its technology-based economic growth. On the other hand, it was generalized to classify countries according to economic competitiveness since Poter's theoretical contribution [13]. Erkut [14] classified countries according to their stage of competitive advantage (factor-driven, efficiency-driven, innovation-driven) by using an expert survey on entrepreneurship and innovation. In an innovation-driven economy, technology is considered as one of its most important elements [14].

The NIS refers to the network of institutions in the public and private sectors interacting with all national organizations and institutions [8] related to science and technology, and R and D activities [7]. Analysis of the R and D mechanism for national R and D programs may be understood in the context of the roles of their respective elements in the single framework of the NIS.

As shown in Figure 1, the mechanisms for managing and adjusting national R and D programs under the NIS may be classified at the panministerial level, as well as at the levels of ministries pertaining to science and technology, supporting organizations, and industry/university-research institutions; supporting organizations herein refer to R and D management agencies, which receive their R and D budget from the government, and manage and execute them [2].

Among the KRW19.4 trillion 2017 budget for national R and D programs, the major R and D budget was approximately KRW12.0049 trillion. In total, 34 government ministries proceed with national R and D programs, and the budget for such R and D projects is on the rise every [15].

Most goverrnment ministries are executing R and D projects and thus have supporting organizations under their umbrellas (R and D management agencies) delegate their respective R and D budgets and projects.

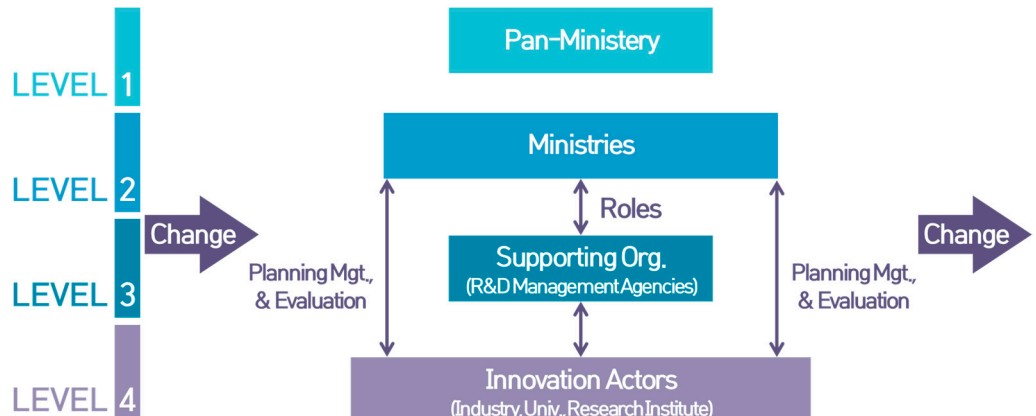

**Figure 1.** Mechanism of the national R and D program management in the National Innovation System. Source: Impact of Organizational Competencies on the Performance of R and D Management Agencies in Korea [2].

As a result, the co-operation and adjustment of governance on science and technology policies and research-management mechanisms have increasing importance, and adjustments in the relationships between participants and strategic-planning capabilities are required.

The adjustment mechanisms in the aspects of research planning and management under the NIS may be classified into a horizontal adjustment mechanism, a mechanism for executing and allocating research expenses, and a mechanism for industry–academia co-operation. Consideration should also be given to a mechanism for panministerial participation and civic participation (by experts and civic organizations) in such adjustments, and private firms' participation in science and technology investment [2].

Thus, procedures for bringing about co-ordination and consensus between the interested parties are essential for successful policy-making decisions and, as shown in Figure 2, such decisions are accompanied by co-ordination, planning, and management made at the levels of high-ranking policymakers, middle-ranking project managers and evaluators, and project performers according to the policy (macro), project (meso), and task (micro) levels [2,16]. In this case, R and D management agencies intervene as an element in the NIS at the mesolevel of project managers and evaluators.

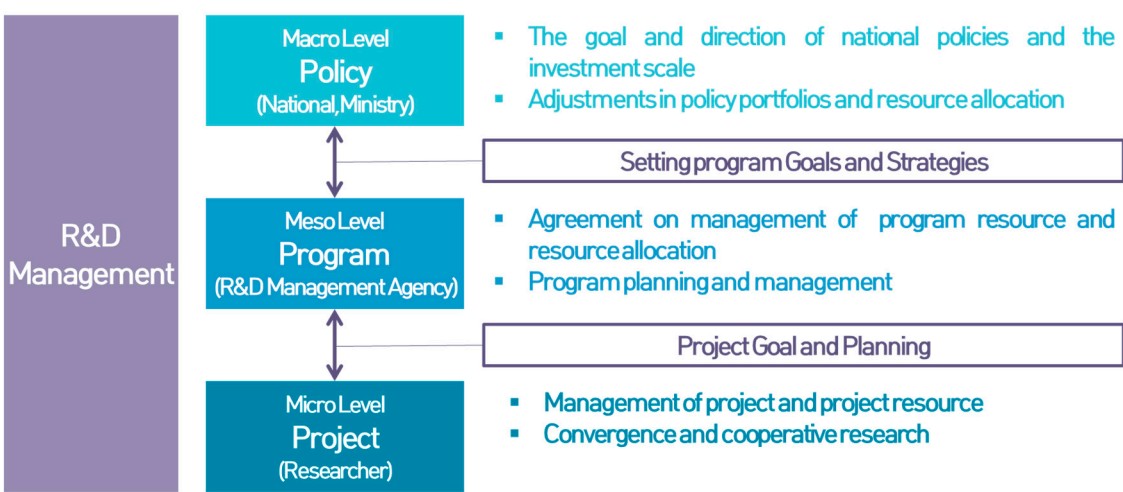

**Figure 2.** Levels of managing science and technology policies, programs, and projects. Source: [2].

*2.2. Overview of Public R and D Management Agencies*

First of all, the ground law for R and D management agencies is described below. They are grounded on Article 11(4) of the Framework Act on Science and Technology [17] and Article 2(6) of the Regulations on the Management etc. of National Research and Development Programs [18]. The grounds for their establishment are separately prescribed in laws and regulations applicable to each ministry.

The purpose of R and D management agencies is originally to overcome restrictions of the Korean administrative system focused on the nurture of generalists, and to secure expertise in the planning, management, and assessment of national R and D programs. Moreover, these institutions are mainly concentrated on assuming the role of inducing competition among researchers in order to enhance research efficiency, and performing micromanagement at the project level.

As mentioned in the introduction, the current system of R and D management agencies can be seen as the result of the advancement of public institutions in 2008. First, at the time, the Ministry of Economy and Finance finally determined examination targets among R and D management agencies as those in cases, where there is inefficiency since organizations exist simultaneously to achieve the same policy objective, and second, where the creation of a new added value is expected as a synergy effect of organizational integration because similar organizations exist in the same sector. The directions of overhauling such institutions are to restructure overlapped and similar organizations into a single integrated one by merging them, to streamline managerial organizations and noncore programs according to the above merger, and to realign functions and personnel on the basis of originally targeted projects.

Such specific examples in education, science, technology, and industry are as follows. The Ministry of Education and the Ministry of Science and Technology integrated three existing organizations into the single one of the National Research Foundation of Korea (NRF) on the basis of the principle of one R and D management agency within one ministry. In the process of such combination, the Ministry of Knowledge Economy had to maintain the function of the organization responsible for technology commercialization and did not combine its organizations under the principle of one R and D management agency within one ministry in view of the distinct characteristics of the Korea Institute of Energy Technology Evaluation and Planning (KETEP). Instead, it integrated six existing organizations into three ones: the Korea Evaluation Institute of Industrial Technology (KEIT), the Korea Institute of Energy Technology Evaluation and Planning (KETEP), and the Korea Institute for Advancement of Technology (KIAT).

As noted in the introduction, as of now, 12 central administrative organizations operate 17 R and D management agencies and entrust them with tasks including the planning, evaluation, and management of R and D projects that fall under their jurisdiction. These 17 agencies are classified into two types according to their R and D management duties, as seen in Table 1.

The two above types are subdivided into eight organizations whose main purpose is the management of R and D projects, and nine ones that regard the management of R and D projects as one of their supplementary tasks.

Second, reviewing R and D management agencies in terms of their R and D cost-management scale shows, as seen in Table 2, the 17 agencies performed the planning, management, and evaluation of R and D projects with a total of KRW10.7 trillion as of 2017, and the ratio of the planning and evaluation costs to the total project cost was 3.1%.

However, the 2017 ratios of the planning and evaluation expenses to the total project cost were, respectively, 7.4% for the Korea Institute of Planning and Evaluation for Technology in Food, Agriculture, and Forestry, 6.2% for the Korea Creative Content Agency (Kocca), 1.6% for the NRF, and 2.3% for the Institute for Information and Communications Technology Promotion (IITP), which shows that planning and assessment costs are distributed among organizations without common standards and principles for such an allotment.

**Table 1.** Current status of R and D management agencies (2018).

| Type | Number of Agencies | R and D Management Agency |
|------|--------------------|---------------------------|
| Main Purpose (Mgt. of R and D Projects) | 8 EA | National Research Foundation of Korea (NRF), Institute for Information and Communications Technology Promotion (IITP), Korea Evaluation Institute of Industrial Technology (KEIT), Korea Institute of Energy Technology Evaluation and Planning (KETEP), Korea Technology and Information Promotion Agency for SMEs (TIPA), Korea Institute of Marine Science and Technology Promotion (KIMST), Korea Institute of Planning and Evaluation for Technology in Food, Agriculture and Forestry (IPET), Korea Agency for Infrastructure Technology Advancement (KAIA) |
| Supplementary Tasks (Mgt. of R and D Projects) | 9 EA | Korea Institute for Advancement of Technology (KIAT) National IT Industry Promotion Agency (NIPA) Korea Health Industry Development Institute (KHIDI) Korea Environmental Industry and Technology Institute (KEITI) Korea Meteorological InstituteKorea Creative Content Agency (KOCCA) Korea Sports Promotion FoundationKorea Foundation of Nuclear Safety Korea Forestry Promotion Institute (KOFPI) |

Source: [19].

**Table 2.** Current status of R and D management-agency budgets in 2017 (Unit: KRW million, %).

| Classification | Total | Program and Project Costs (A) | Planning and Evaluation Costs (B) | Administrative Expense | | Ratio of the Planning and Evaluation Costs to the Total Project Costs (B/A) |
|----------------|-------|-------------------------------|-----------------------------------|-------------|------------------|----------------------------------------------------------------------------|
| | | | | Direct Costs | Employment Costs | |
| **R and D** | 106,657 | 103,432 | 3227 | 1179 | 2048 | 3.1 |

Source: [19]. Note: Planning, evaluation, and management costs directly related to the management of programs and projects were set as direct costs.

## 2.3. Research Methods

As the role of science and technology becomes significant in securing the nation's competitiveness, and as the R and D budget accordingly increases, the need for a life-cycle system for the efficient management of national R and D is growing [3].

This study analyzes the status of the operational efficiency of R and D management agencies from a macro perspective concerning the life-cycle aspect of R and D involving the process of Planning (Plan), Management (Do) and Evaluation (See). In this study, the major functions of R and D management agencies were classified into the life-cycle of R and D project. The hypotheses were established on the assumption that there would be difference in the perception of staff members on the detailed function.

**H1:** *There is a difference in the perception of staff members according to the detailed function of the Strategy and Planning of R and D management agencies.*

**H2:** *There is a difference in the perception of staff members according to the detailed function of the Management and Support of R and D management agencies.*

**H3:** *There is a difference in the perception of staff members according to the detailed function of the Evaluation and Utilization of R and D management agencies.*

Based on the analysis results of this study, we also intended to present possible ways of improvement for the operational efficiency of these agencies.

For data collection, we carried out a survey of 117 people from 1 July 2017 until 15 August 2017, using a sample of seven to 10 staff members in the 17 agencies responsible for national R and D management. In addition, this study referred to the results of an unstandardized interview held during the above survey period.

The survey was analyzed on the basis of each measurement result (five-point Likert scale) of each questionnaire category. To test the hypothesis, and to prove the statistical validity of the measurement result, a one-sample t-test was carried out.

## 3. Analysis of Current Status of Public R and D Management Agencies

### 3.1. Governance System of R and D Management Agencies

Table 3 showed the results of the perception difference on governance system of R and D management agencies.

**Table 3.** Governance system of R and D management agencies.

| Questionnaire | One-Sample t-test | | | |
|---|---|---|---|---|
| | Average Difference | Standard Deviation | t | p-Value |
| Adequacy of the number of R and D management agencies | −0.1727 | 0.9941 | −1.822 | 0.071 * |
| Adequacy of systematic support for R and D management agencies | 0.1455 | 0.8866 | 1.721 | 0.088 * |

Source: Data analyzed by the authors. Notes: * sig (p-value) < 0.1. One-sample t-test indicated that the response was statistically positive.

First, the results of questionnaire responses on the adequacy of the total number of R and D management agencies that operate at the national level show that the average score is 2.83, and that the responses of strongly agree (2%) and agree (27%) are 29%, and the responses of disagree (30%) and strongly disagree (9%) are 39%. These findings illustrate that positive opinions on the current system of R and D management agencies in terms of the number thereof are even less than half of the whole responses, and suggest that there is a need for in-depth discussion on the proper number of R and D management agencies that is helpful for them to better serve their functions and roles.

Second, according to the results of questionnaire responses on the adequacy of the applicable laws and regulations and systematic support for R and D management agencies, the average score is 3.15, the responses of strongly agree (3%) and agree (38%) are 41%, and the responses of disagree (27%) and strongly disagree (1%) are 28%. This indicates that the positive perceptions of legal and systematic government support for R and D management agencies are higher than the negative perceptions.

### 3.2. Analysis of Current Status from the R and D Life-Cycle Perspective

The major functions of R and D management agencies are classified into aspects of strategy and planning, management and support, evaluation and utilization, and the results of the hypothesis test about perceptions of their specific details are as Table 4.

First, the hypothesis about the strategy and planning function of R and D management agencies *(H1)* was examined. The hypothesis test results were partially adopted. And showed that, in terms of the strategy and planning function of R and D management agencies, there are difference in perception about the agency performance strategy and planning functions, adequacy of personnel acquisition for strategy and planning and Functional co-operation between R and D management agencies in strategy and planning. Positive perceptions are higher that the adequacy of the strategy and planning performance of these agencies and the acquisition of personnel specialized in such strategy and planning. But negative perceptions of co-operation among such agencies are higher.

Second, the results of the hypothesis test for the management and support functions of R and D management agencies *(H2)* were partially adopted. In terms of the management and support functions of R and D management agencies, positive perceptions of their efforts to minimize researchers' administrative burdens are higher.

**Table 4.** Survey results about the major functions of R and D management agencies.

| Questionnaire | | One-Sample t-test | | | |
| | | Average Difference | Standard Deviation | t | p-Value |
|---|---|---|---|---|---|
| Strategy and Planning | 1. Agency performance strategy and planning functions | 0.3419 | 0.8112 | 4.559 | 0.000 *** |
| | 2. Adequacy of personnel acquisition for strategy and planning | 0.3077 | 0.7819 | 4.256 | 0.000 *** |
| | 3. Establishment of clear roles along with ministries in the strategy and planning process | 0.0862 | 0.9831 | 0.944 | 0.347 |
| | 4. Budget support for strategy- and planning-function performance | 0.0769 | 0.8321 | 1.000 | 0.319 |
| | 5. Sufficient organization operation for strategy and planning | 0.0086 | 0.8797 | 0.106 | 0.916 |
| | 6. Functional co-operation between R and D management agencies in strategy and planning | −0.2991 | 0.9122 | −3.547 | 0.001 *** |
| Management and Support | 7. Adequacy of the scale of budget support for R and D management | 0.0855 | 0.9245 | 1.000 | 0.319 |
| | 8. Sufficiency of R and D management personnel | −0.3333 | 0.9469 | −3.808 | 0.000 *** |
| | 9. Autonomy in R and D management | −0.1293 | 0.9281 | −1.501 | 0.136 |
| | 10. Autonomy in budget formulation and execution | −0.0256 | 0.9141 | −0.303 | 0.762 |
| | 11. Information- and resource-sharing among participants in programs and projects | 0.0513 | 0.7970 | 0.696 | 0.488 |
| | 12. Efforts to minimize researchers' administrative burdens | 0.2991 | 0.8834 | 3.663 | 0.000 *** |
| Evaluation and Utilization | 13. Expertise acquisition in the evaluation of R and D projects | 0.6154 | 0.8184 | 8.134 | 0.000 *** |
| | 14. Establishment of the system for evaluating R and D projects | 0.7607 | 0.8575 | 9.595 | 0.000 *** |
| | 15. Construction of the feedback system on evaluation results | 0.2393 | 0.8372 | 3.092 | 0.002 *** |
| | 16. Sharing project achievements and database management | 0.3846 | 0.9083 | 4.580 | 0.000 *** |
| | 17. System establishment and operation for sharing achievements among related organizations | −0.1538 | 0.8672 | −1.919 | 0.057 * |
| | 18. Retention and utilization of personnel specialized in evaluation | 0.4872 | 0.8572 | 6.148 | 0.000 *** |
| | 19 Systematic management of achievements and follow-up management | 0.2308 | 0.7811 | 3.196 | 0.002 *** |

Source: Data analyzed by the authors. Notes: * sig (p-value) < 0.1, *** sig (p-value) < 0.01. One-sample t-test indicated that the response was statistically positive.

Third, the results of the evaluation and utilization functions of R and D management agencies *(H3)* were all adopted at 10% significance level. In terms of the evaluation and utilization functions of R and D management agencies, positive perceptions of the establishment of the scheme and system for evaluating R and D projects, the acquisition of expertise in the assessment of R and D projects, the retention and utilization of personnel specialized in evaluation, the sharing of project achievements, research tasks, and database management, the construction of the system for giving feedback on evaluation results, and the systematic management of achievements and their follow-up management are higher, while negative perceptions of the establishment and operation of the system for sharing achievements among related organizations are higher.

As illustrated above, it was found that, among the major functions of R and D management agencies, those that turned out to be lacking and need to be improved in the future are functional co-operation among these agencies in terms of strategy and planning, the sufficiency of R and D management personnel in terms of management and support, and the establishment and operation of the system for sharing achievements among related organizations in term of evaluation and utilization.

## 3.3. Current Status of Project-Management Systems (PMS)

PMS refers to a system that computerizes R and D management agencies' tasks (including planning, agreement, evaluation, performance, and follow-up management) and their history information to effectively undertake administrative management and support, including the announcement, application, selection, and assessment of government R and D program and projects.

The main function of the system is to carry out a role of providing direct on-line support to researchers and R and D management agencies throughout their life-cycle processes.

As of now, there are a total of 151 regulations and guidelines that vary across R and D management agencies, and 20 different systems for managing projects are in operation by 17 ministries, as seen in Table 5 below. It was found that these agencies use 4.7 systems on average for annual funding, whereas universities use 8.2 systems on average [20].

**Table 5.** Current status of project-management systems (PMS).

| No. | Ministry | R and D Management Agency | Project mgt. System (PMS) | Supplementary System (Employment, Performance Mgt., etc.) |
|---|---|---|---|---|
| 1 | Ministry of Education (MOE) | National Research Foundation of Korea (NRF) | e-R and D, Brain Korea 21 plus comprehensive information mgt. system | Korean Researcher Information (KRI), Sunggwamaru System |
| 2 | Ministry of Science and ICT(MSIT) | National Research Foundation of Korea (NRF) | e-R and D | |
| 3 | | Institute for Information and Communications Technology Promotion (IITP) | EZOne | ICT-Bay |
| 4 | Ministry of SMEs and Startups (MSS) | Korea Technology and Information Promotion Agency for SMEs (TIPA) | Small and Medium Business Technology Development Mgt. System (SMTECH) | |
| 5 | Ministry of Culture, Sports, and Tourism (MCST) | Korea Creative Content Agency (KOCCA) | CTRD | |
| 6 | Ministry of Health and Welfare (MOHW) | Korea Health Industry Development Institute (KHIDI) | Htdream | |
| 7 | Ministry of Environment (ME) | Korea Environmental Industry and Technology Institute (KEITI) | Eco-PLUS | |
| 8 | Ministry of Oceans and Fisheries (MOF) | Korea Institute of Marine Science and Technology Promotion (KIMST) | Integrated R and D mgt. system | |
| 9 | Ministry of Land, Intrastructure and Transport (MOLIT) | Korea Agency for Infrastructure Technology Advancement (KAIA) | Infrastructure R and D mgt. System | |
| 10 | Ministry of Agriculture, Food, and Rural Affairs (MAFRA) | Korea Institute of Planning and Evaluation for Technology in Food, Agriculture and Forestry (IPET) | Food, Agriculture, and Forestry R and D Information Service (FRIS) | |
| 11 | Ministry of Trade, Industry, and Energy(MOTIE) | Korea Evaluation Institute of Industrial Technology (KEIT) | iTECH+ | |

**Table 5.** *Cont.*

| No. | Ministry | R and D Management Agency | Project mgt. System (PMS) | Supplementary System (Employment, Performance Mgt., etc.) |
|---|---|---|---|---|
| 12 | | Korea Institute of Advancement of Technology (KIAT) | K-PASS | |
| 13 | | Korea Institute of Energy Technology Evaluation and Planning (KETEP) | Green Energy New Innovative Expert (GENIE) | IPM |
| 14 | Ministry of the Interior and Safety (MOIS) | National Disaster Management Research Institute (NDMI) | R and D Project Management System | |
| 15 | Ministry of Food and Drug Safety (MFDS) | National Institute of Food- and Drug-Safety Evaluation (NIFDS) | R and D Mgt. System | |
| 16 | Culture Heritage Administration (CHA) | National Research Institute of Cultural Heritage (NRICH) | R and D Project Mgt. System | |
| 17 | Defense Acquisition Program Administration (DAPA) | Defense Agency for Technology and Quality (DTAQ) | Defense Technology Information Service (DTiMS) | |
| 18 | Korea Meteorological Administration (KMA) | Korea Meteorological Institute (KMI) | R and D Mgt. System for KMA | Industrial Open Market for Weather and Climate |
| 19 | Korea Forest Service | Korea Forestry Promotion Institute (KOFPI) | Forestry, Science, and Technology Information Service (FTIS) | |
| 20 | Rural Development Administration (RDA) | Rural Development Administration (RDA) | Agriculture science Technology Information System (ATIS) | |

Source: [20].

In terms of researchers, regulations, and guidelines varying across organizations, separate logs into different systems, overlapped recording and management of materials, and disparate services are causing much inconvenience to researchers. In terms of managers, insufficient overall panministerial management, restrictions on real-time information sharing, limitations on co-operation between organizations, deepening differences on service levels, and budgetary waste and inefficiency are causing to managers, and therefore, it is required to standardize regulations and guidelines for R and D management and integrate and overhaul the relevant systems.

## 4. Discussion: Methods for Improving the Operational Efficiency of Public R and D Management Agencies

In the following section, we propose improvements to complement the overall weaknesses of the operational efficiency of public R and D management agencies.

### 4.1. Unification of R and D Planning, and Evaluation Functions by Individual Ministries

Analysis of the survey finds that there are limits to the enhancement of the efficiency of R and D investment due to the system of operating a large number of R and D management agencies. As a counterplan to this, we to propose that overhauling these agencies under a new system of a single R and D management agency within a ministry, and advancing the effect of economies of scale and the efficiency of research management.

As shown in Table 6 below, the above proposal shows a positive response rate of 41.7% and a negative response rate of 22.6% about the survey item of "functional co-ordination and unification among R and D management agencies."

**Table 6.** Functional co-ordination and unification among R and D management agencies.

| Questionnaire | One-Sample T-Test | | | |
| --- | --- | --- | --- | --- |
| | Average Difference | Standard Deviation | t | p-Value |
| Functional Co-Ordination and Unification among R and D Management Agencies | 0.3652 | 1.1648 | 3.363 | 0.001 *** |

Source: Data analyzed by the authors. Notes: *** sig (p-value) < 0.01. One-sample t-test indicated that the response was statistically positive.

To be more specific, the R and D planning and evaluation functions of 17 R and D management agencies would be unified by individual ministries, which is needed to strengthen sectoral co-ordination and co-operation. Considering the adequacy of the R and D management scale, expertise in such management, and the acquisition of the collaboration framework as well as the unification of the above functions by each ministry, 17 R and D management agencies within 12 ministries and offices shall be combined into 10 agencies within 10 ministries. As demonstrated in Table 7 below, the reorganization of 12 R and D management agencies within seven ministries (including the Ministry of Science and ICT; the Ministry of Trade, Industry, and Energy; the Ministry of Culture, Sports, and Tourism; the Ministry of Environment/the Korea Meteorological Administration; and the Ministry of Agriculture and Forestry/the Korea Forest Service) subject to combination into five agencies within five ministries is a key to the above combination.

**Table 7.** Method of integrating R and D management agencies per ministry.

| Ministry | As-Is | To-Be |
| --- | --- | --- |
| Ministry of Science and ICT(MSIT) | National Research Foundation of Korea (NRF), National IT Industry Promotion Agency (NIPA), Institute for Information and Communications Technology Promotion (IITP) | One Integrating R and D Management Agency (MSIT) |
| SMinistry of Trade, Industry and Energy (MOTIE) | Korea Evaluation Institute of Industrial Technology (KEIT), Korea Institute of Energy Technology Evaluation and Planning (KETEP), Korea Institute for Advancement of Technology (KIAT) | One Integrating R and D Management Agency (MOTIE) |
| Ministry of Culture, Sports and Tourism (MCST) | Korea Creative Content Agency (KOCCA), Korea Sports Promotion Foundation | One Integrating R and D Management Agency (MCST) |
| Ministry of Environment (ME)+Korea Meteorological Administration (KMA) | Korea Environmental Industry and Technology Institute (KEITI), Korea Meteorological Institute | One Integrating R and D Management Agency (ME) |
| Ministry for Food, Agriculture, Forestry and Fisheries (MIFAFF)+Korea Forest Service(KFS) | Korea Institute of Planning and Evaluation for Technology in Food, Agriculture and Forestry (IPET), Korea Forestry Promotion Institute (KOFPI) | One Integrating R and D Management Agency (MIFAFF) |

Source: Data prepared by the authors.

Furthermore, common guidelines would be formulated in order to prevent the establishment of too many R and D management agencies by each individual ministry, which is required to make constant adjustments to the functions of such agencies by eliminating incompetent organizations.

*4.2. Establishment of Panministerial Management System of R and D Management Agencies*

It is necessary to establish a panministerial management system of R and D management agencies in order to not only prevent such organizations from springing up across ministries, but also to allow them to have certain levels of capabilities, rather than having them make one-off innovations.

To this end, it is first urgently needed to create legal grounds for common principles on the definition, establishment, and requirements for the designation, functions, roles, and evaluation of R and D management agencies. The names of such agencies, whose functions have been completely overhauled, within the respective ministries would be specified in applicable laws, and the above panministerial system would be managed to ensure that only designated R and D management agencies are allowed to carry out R and D planning, management, and evaluation functions. On top of this, the respective ministries' ground laws for the incorporation of these agencies and the Act on the Management of Public Institutions would be amended.

Second, we propose that the Office of Science and Technology Innovation within the Ministry of Science and ICT devise a system for the evaluation of R and D management agencies' competencies and performance levels, and that in-depth evaluations of all R and D management agencies be made. Specific methods of such assessments are to organize an evaluation panel and to examine the reflection of the evaluation manual and achievements. Major evaluation indicators include expertise in planning and evaluation, competencies, the quality of research and administrative services, and the application of R and D in institutional improvements. The results of the above evaluations would first be reflected in the determination of planning and evaluation costs for each ministry, and later be used in the transfer and overhaul of the functions of incompetent R and D management agencies.

As seen in Table 8 below, the preceding proposal shows a positive response rate of 67.8% and a negative response rate of 4.4% about the need for performance management and the establishment of each evaluation system considering organizational characteristics.

**Table 8.** Performance management and the establishment of each evaluation system considering organizational characteristics.

| Questionnaire | One-Sample T-Test | | | |
| --- | --- | --- | --- | --- |
| | Average Difference | Standard Deviation | t | p-Value |
| Performance Management and the Establishment of Each Evaluation System Considering Organizational Characteristics | 0.8696 | 0.8535 | 10.926 | 0.000 *** |

Notes: *** sig (p-value) < 0.01. One-sample t-test indicated that the response was statistically positive.

*4.3. Improvement of R and D Development Agencies' Expertise and R and D Management Services*

We propose overhauling these agencies' functions with a focus on the realization of researcher-centered research environments and the enhancement of R and D investment efficiency, as well as making the following systematic improvements for the advancement of expertise in R and D planning, management, and evaluation.

4.3.1. Standardising R and D Regulations of Individual Ministries

We suggest streamlining and standardizing the relevant regulations, including those on principles to push for government R and D, the procedures thereof, the methods of managing such R and D, sanctions thereon, and institutionalizing improvements in applicable R and D laws and regulations through a collection of researcher voices.

Researchers have raised a lot of complaints since R and D management agencies respectively apply different standards for their management and principles:

"As you know, we have to comply with the Ministry of Science and ICT's management standards, principles, and specific guidelines, and the operational rules of the respective ministries. However, such rules are so specific that they are too strict and intrusive from the standpoint of researchers. We have no other choice but to more strictly apply these rules in a situation where we are under audit over and over again, even though we do not want to do so. If there is a problem, we are forced to make new guidelines to solve it" (☆☆☆ Agency, Team Leader △△△△△; interview with the R and D management agency in August 2017).

The above interview results are seen as an example of the need for the standardization of R and D regulations by each ministry.

### 4.3.2. Improvement and Advancement of PMS

We suggest that PMS varying across organizations be integrated into one system in phases so as to provide streamlined services.

As shown in Table 9, there was a positive response rate of 67.8% and a negative response rate of 7.8% about the survey item of the need for "information- and resource-sharing between R and D management agencies".

**Table 9.** Information- and resource-sharing between R and D management agencies.

| Questionnaire | One-Sample t-Test | | | |
| --- | --- | --- | --- | --- |
| | Average Difference | Standard Deviation | t | p-Value |
| Information and Resource Sharing Among R and D Management Agencies | 0.8000 | 0.8502 | 10.091 | 0.000 *** |

Notes: *** sig (p-value) < 0.01. One-sample t-test indicated that the response was statistically positive.

Specifically, with respect to the standardization of PMSs (for the planning, selection, evaluation, and performance management), ministries would launch their efforts to standardize their respectivly operating PMSs and draw up ways of standardizing them. In addition, ministries need to share information on research projects (including planning, evaluation, field of technology, researchers) in real time so that they can employ such information in planning new projects, promoting expertise in evaluating, enabling co-operation among ministries, and formulating effective R and D investment strategies.

### 4.3.3. Reorganization of the System of R and D Planning and Evaluation Costs

R and D planning and evaluation costs refer to those spent on the planning, evaluation, and management of R and D projects. The problem with such costs, however, is that they are spent in operating organizations as personnel and overhead costs, rather than being used for their original purposes, thereby restricting the specialized planning and management of R and D.

Therefore, we propose that the distributed budgets, as shown in Table 10 below, be unified as those for projects funded by each ministry. Specifically, dispersed personnel, overhead, and planning and evaluation costs across projects would be formulated and managed by the respective ministries, and standards for calculating planning and evaluation expenses would be created in light of not only the number of tasks, evaluation methods, and evaluations, but also the proper size of such costs in order to strengthen expertise in the utilization of planning and research achievements. Moreover, the evaluation results of R and D management agencies would be linked to and differently reflected in budget distribution for planning and evaluation costs.

**Table 10.** Comparison between before and after the reorganization of the system of R and D planning and evaluation costs.

| | Current System | Reorganized System |
|---|---|---|
| Budget Structure | Employment and overhead costs (major R and D) | Block Funding (Employment + overhead + planning and evaluation costs) |
| | Planning and evaluation costs (major R and D) | |
| | Employment and overhead costs (ordinary R and D) | |
| | Planning and evaluation costs (ordinary R and D) | |
| | Exclusive planning and evaluation costs | |

Source: Data prepared by the authors.

In regard to this problem, those in charge of R and D management agencies say:

"As the ratio of government funds to operating costs is low, we fund about 90% of operating costs through research planning and evaluation costs. The problem with this is that since most of operating costs are financed by research planning and evaluation costs, the funding of operating costs is not stable. This could be a stumbling block to the stable operation of our organization such as the formulation of a project plan for the next year or personnel operation" (☆☆☆ Agency, △△△△△ A person of accounting; interview with the R and D management agency in August 2017).

The above interview results are seen as an example of the need for supplementing the existing distribution of planning and evaluation costs.

### 4.3.4. Development of a Supporting System to Boost Expertise in Planning and Evaluation

First, in an effort to boost expertise in planning and evaluation, we propose that planning managers (PM) mainly increase in industrial fields and markets, and that the status of planning managers be enhanced within R and D management agencies.

Second, we propose that a common pool of evaluators be created, and the respective R and D management agencies' evaluation-personnel databases be linked to their systems and used jointly by such agencies in order to bolster a panministerial sharing system of researcher achievements. Lastly, we propose that an association of R and D management for strengthening mutual linkage and co-operation be formed so as to facilitate R and D co-operation between R and D management agencies and the expansion of convergent and comprehensive R and D, and that the government provide support to this.

## 5. Conclusion and Further Research

This study has analyzed the actual conditions of R and D management agencies from the perspective of an R and D life-cycle process, and has presented improvement methods for the operational efficiency of these agencies.

In this study, the major functions of R and D management agencies are classified into the life cycle of R and D projects. The hypotheses were established on the assumption that there would be difference in the perception of staff members on the detailed function. To do this, we conducted a questionnaire survey of staff members in the 17 R and D management agencies. Hypothesis testing was conducted using the collected data.

Analysis of the current status of R and D management agencies found that they should be restructured into organizations of appropriate numbers as related to their governance systems, however, legal and systematic of the government support turns out to be positive. In addition, there is a need for functional co-operation between these agencies in terms of strategy and planning, the sufficiency of research management personnel in terms of management and support, and the development and operation of a shared system of achievement between the related organizations in terms of evaluation and utilization. Additionally, regulations and guidelines should be standardized with respect to the current PMS status, and that PMS is needed to integrate and overhaul systems.

Based on the analysis results, we offer suggestions in three areas for improvement: (a) unification of R and D planning and evaluation of individual ministries; (b) establishment of a panministerial management system for public R and D management agencies; (c) improvement and development of public R and D agencies' expertise and management services.

Lastly, the above analysis was also measured as individual performance recognition at R and D management agencies rather than simply depending on quantitative data regarding performance because standardized classification of such performance has not yet been made in connection with the future direction of this study. This study is supplemented with a detailed plan for investigation. If studies, such as efficiency analysis or other performance analysis, are done in the next three to five years, better methods of improvement for the operational efficiency of R and D management agencies can be presented.

**Author Contributions:** B.Y.H. developed the concept and wrote the paper. E.S.B. performed the statistical analyses. H.D.H. collected the data and analyzed policy trends. D.C.K. designed the research and wrote the paper. All authors read and approved the final manuscript.

**Funding:** This paper was supported by KISTEP Internal Project 'A Research on S and T regulatory reform' in the publishing fee.

**Conflicts of Interest:** The authors declare no conflict of interest.

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
