# Peer review of "Operational-Efficiency Improvement of Public R and D Management Agencies in South Korea"

_2199-8531, doi:10.3390/joitmc5010013_

Round 1

Reviewer 1 Report

The work entitled: A Study on the Improvement of Operational

Efficiency of Public R & D Management Agencies in South Korea. Its main objective is to suggest ways to improve the operational efficiency of public R & D management agencies based on analysis of their current status.

- The work is correctly written and presented, it is easy to read, the methodology used is correct and adequate.

- The work presents, based on the results of the analysis, suggestions for improvement in three areas and adequately discusses the possible improvements and limits of this study.

Suggestions for improvement for authors:

1/ Reduce the title, title proposed: "Improvement of Operational Efficiency of Public R & D Management Agencies in South Korea"

2/ Improve layout of tables and figures: Figure 1 and 2.

3/ line 52: National Innovation System(NIS)?;

Congratulated the authors for the work done

Author Response

I would like to submit the revising report and revise marked full paper.

Hope it would do.

Please see the below and attached.

Reviewer 1

Reviewee

1. Reduce the   title

Revised

“Improvement   of Operational Efficiency of Public R & D Management Agencies in South   Korea”

2.   Improve layout of tables and figures: Figure 1 and 2.

The figure   1,2 and table 2 have been replaced

3. line 52: National Innovation System(NIS)?

Please   refer to 2.1 where NIS has been described more concretely

 Sincerely Yours,

Reviewer 2 Report

This study on the improvement of operational efficiency of public R&D management agencies in South Korea is a very interesting case study regarding public R&D management, with interesting results and conclusions.

Nevertheless, I believe that in this form the article resembles more of a technical report or a "measurement without theory".

First of all, I would expect the authors to include hypotheses, which is missing. Second, the motivation seems to be not justified. For example, in lines 34-37 the authors claim that there is a decline in the efficiency of government R&D and that there is criticism - but do not cite any resources.

Third, I believe that the research background is not including the correct references. Citing a historical case study about Roman era for R&D management is not appropriate. I would expect the authors to focus more on the competitiveness literature around Porter (1990). For example, Erkut's both approaches (see below) support the authors' point of view that a single agency would ease innovations and entrepreneurial activities and explain it from a theoretical view, allowing also empirical evidence to support it:

Erkut, B. (2016a). "Structural Similarities of Economies for Innovation and Competitiveness - A Decision Tree Based Approach", SOEP, DOI:10.18559/SOEP.2016.5.6

Erkut, B. (2016b). "Entrepreneurship and Economic Freedom – Do Objective and Subjective Data Reflect the Same Tendencies?", EBER, DOI:10.15678/EBER.2016.040302

Fourth, I believe that the quality of graphics is way too low - these need to be remade by e.g. Photoshop.

I wish the authors all the best with their article and hope that these comments are valuable for their research.

Author Response

I would like to submit the revising report and revise marked full paper.

Hope it would do.

Please see the below and attached.

Reviewer 2

Reviewee

1. I would expect the authors to   include hypotheses.

Hypotheses   were established and the results were summarized

2. In   lines 34-37 the authors claim that there is a decline in the efficiency of   government R&D and that there is criticism - but do not cite any   resources.

Source:   Yang, et al. 2015; Hwang et al. 2018; Hong et al. 2018

3. I   believe that the research background is not including the correct references.   Citing a historical case study about Roman era for R&D management is not   appropriate. I would expect the authors to focus more on the competitiveness   literature around Porter (1990). For example, Erkut's both approaches (see   below) support the authors' point of view that a single agency would ease   innovations and entrepreneurial activities and explain it from a theoretical   view, allowing also empirical evidence to support it:

Theoretical   background and refernces were added

4. , I   believe that the quality of graphics is way too low - these need to be remade   by e.g. Photoshop.

The figure 1,2 and table 2 have been replaced

 Sincerely Yours,

Round 2

Reviewer 2 Report

Dear Authors,

thank you for integrating the suggested changes and for the interesting article. I believe this will be a good contribution, both in terms of the conceptual direction passing the "test of time" and in terms of its policy content. Good luck with your future work!